# Effects of physical, chemical, and biological ageing on the mineralization of pine wood biochar by a *Streptomyces* isolate

**Nayela Zeba**[1], **Timothy D. Berry**[1], **Kevin Panke-Buisse**[2], **Thea Whitman**[1]*

**1** Department of Soil Science, University of Wisconsin-Madison, Madison, Wisconsin, United States of America, **2** Dairy Forage Research Center, University of Wisconsin-Madison, Madison, Wisconsin, United States of America

* twhitman@wisc.edu

**Data Availability Statement:** Data are deposited in the DOE ESS-DIVE repository under: https://data.ess-dive.lbl.gov/datasets/doi:10.15485/1856070 (DOI: 10.15485/1856070).

## Abstract

If biochar is to be used for carbon (C) management, we must understand how weathering or ageing affects biochar C mineralization. Here, we incubated aged and unaged eastern white pine wood biochar produced at 350 and 550°C with a *Streptomyces* isolate, a putative biochar-decomposing microbe. Ageing was accelerated via three different processes, namely, (a) physical ageing–subjecting biochar to alternating freeze-thaw and wet-dry cycles, (b) chemical ageing–treating biochar with concentrated hydrogen peroxide and (c) biological ageing–incubating biochar in the presence of nutrients and microorganisms. Elemental composition and surface chemistry (Fourier Transform Infrared spectroscopy) of biochar samples were compared before and after ageing. Biochar C mineralization between ageing treatments was significantly different in the case of 350°C biochar ($p$ value = 0.03). Among the 350°C biochars, physical ageing resulted in the greatest increase (by 103%) in biochar C mineralization ($p$ value = 0.05). However, in the case of 550°C biochar, ageing did not result in a significant change in biochar C mineralization ($p$ value = 0.40). Biochar C mineralization was positively correlated with an increase in O/C ratio post-ageing ($r_s$ = 0.86, $p$ value = 0.01). In the case of 350°C biochar, surface oxidation during ageing enhanced biochar degradation by the isolate. For 550°C biochar, however, ageing did not significantly increase biochar C mineralization, likely due to high condensed aromatic C content and lower surface oxidation during ageing. The results from our study suggest that low temperature aged biochar is more susceptible to biological degradation by soil microbes. These findings have implications for the use of biochar for long term C storage in soils.

## Introduction

Biochar is the carbon-rich solid product of pyrolysis, the process of heating biomass under oxygen limited conditions [1]. Biochar has the potential to be used as a soil amendment for agricultural management (e.g., to increase water holding capacity and enhance nutrient management) and as a carbon (C) management strategy to help mitigate greenhouse gas emissions

**Funding:** TW: DE-SC0020351 and DE-SC0016365, Department of Energy (DOE), Office of Science, Office of Biological and Environmental Research (BER), https://science.osti.gov/ber/. The funders had no role in study design, data collection and analysis, decision to publish, or preparation of the manuscript.

**Competing interests:** The authors have declared that no competing interests exist.

[2]. Converting waste biomass into biochar can potentially be an effective way to sequester C, since the C contained in biochar is generally more resistant to mineralization compared to the C in the parent biomass [2, 3]. However, the net C effects of biochar application in soil depend heavily on system-specific parameters, particularly what would have happened to the parent biomass had it not been used to produce biochar (e.g., would it have decomposed rapidly, or would it have continued to grow and fix C) [4–7].

The persistence of biochar C in soil can be attributed to its high proportion of condensed aromatic C [2, 8, 9], which has been shown to be resistant to mineralization by both abiotic and biotic processes [10, 11]. Further, biochar, while being rich in C, tends to have a low oxygen (O) and hydrogen (H) content, and low O/C and H/C ratios in biochar have been shown to correlate with biochar persistence in soil [12, 13]. The chemical and physical properties of biochar that affect its persistence are initially determined by the production conditions, such as feedstock and production temperature [14]. But once the biochar is deposited in soil, these properties change over time in a process known as weathering or ageing [15–18]. Natural ageing of biochar in soil is a complex process with multiple relevant mechanisms [15]. We focus on three of these dominant mechanisms over the course of this paper:

a. *Physical ageing*—physical breakdown of biochar, primarily by freeze-thaw cycles and changes in temperature and moisture [15, 19–21]

b. *Chemical ageing*—degradation of biochar through abiotic oxidation upon exposure to various oxidizing agents [22–24]

c. *Biological ageing*—biotic degradation and corresponding physical and chemical modifications of biochar by microbes and other soil organisms [25–29]

Commonly reported effects of biochar ageing include a drop in pH, an increase in O content, and an increase in O-containing functional groups on the surface of aged biochar compared to unaged biochar [18, 30]. This suggests that ageing of biochar, both naturally and artificially, causes changes to its elemental composition and surface chemistry. Furthermore, these changes have been shown to affect properties of biochar such as sorption [19, 31] and cation exchange capacity [32, 33]. However, there is limited information on how these changes will alter the mineralizability of biochar itself. Spokas [34] reported an increase in total C mineralization upon incubation of soil amended with 3 year aged woody biochar, primarily due to chemical oxidation of biochar surfaces. On the other hand, Liu et al. [35] observed lower total C mineralization in soil incubations amended with 6 year aged wheat straw biochar due to loss of easily mineralizable C during ageing. Notably, these studies primarily looked at changes in total C mineralization after addition of aged biochar to soil, while the effect of ageing specifically on the mineralizability of the aged biochar itself by soil microbes has not been fully explored.

Investigating the relationship between physicochemical changes due to ageing and biochar mineralizability is one of the primary tranches of this work. Specifically, the aim of this study is to examine the mineralizability of aged biochar by a specific biochar-degrading microbe from a genus that is common to soils worldwide–a *Streptomyces* isolate [36]. We predicted that the change in mineralization with ageing will depend on whether

i. the ageing process results in loss of easily mineralizable C (as indicated by aliphatic chemical groups), which would lead to lower mineralization, or

ii. an increase in the O content (as indicated by the O/C ratios), which would lead to higher mineralization

## Materials and methods

### Production of biochar

Biochar was produced from eastern white pine wood chips (*Pinus strobus* (L.)) at highest treatment temperatures (HTT) of 350 and 550°C in a modified Fischer Scientific Lindberg/Blue M Moldatherm box furnace (Thermo Fisher Scientific, Waltham, MA, USA) under continuous argon flow (1 L min$^{-1}$) and a peak temperature residence time of 30 min [37]. We chose 350 and 550°C HTTs as they fall within the range of temperatures recorded for naturally produced pyrogenic organic matter (PyOM) during wildfires in addition to being common HTTs for biochar production [38, 39]. Including biochar produced at two different temperatures also allowed us to observe the effect that increasing C content and aromaticity has on the mineralizability of aged biochar [11]. Biochar was ground using a ball mill and sieved to a particle size of <45 μm. We used finely ground biochar to maximize the surface area available for both the ageing process and mineralization of biochar C by the *Streptomyces* isolate during incubation. The full details of biochar production can be found in S1 Appendix.

### Ageing of biochar

Biochar produced at 350 and 550°C was subjected to one of three different ageing processes—physical, chemical and biological. We performed all ageing treatments on single batches of biochar to give us a final set of physically, chemically and biologically aged chars produced at 350°C (350PHY, 350CHEM and 350BIO) and 550°C (550PHY, 550CHEM and 550BIO). A batch each of 350°C unaged biochar (350UN) and 550°C unaged biochar (550UN) acted as controls in our study.

**Physical ageing.** For physical ageing, we subjected biochar samples to 20 freeze-thaw-wet-dry cycles between -80°C and 100°C using pint-sized Mason jars (473.18 mL), building on the method reported by Hale et al. [19]. In addition to the freeze-thaw process used by Hale et al., we included a wet-dry cycle to further accelerate physical breakdown of biochar. Wet-dry cycles at different moisture levels have been used in previous studies to artificially age biochar [18]. While our freezing temperature was close to that used by Hale et. al, we chose to dry our samples at 100°C to ensure that the biochar rapidly and completely dried out between cycles. Quartz sand (Sargent Welch, Buffalo Grove, IL, USA) was used to simulate an inactive soil matrix (80 g with a 5% weight biochar amendment) due to its inability to retain water or nutrients. Ultrapure water was added to the jars containing 4 g of biochar to achieve 40% water holding capacity (WHC). During each cycle, the jars were frozen at -80°C for a median time of 7 hours (min 5 h–max 48 h), thawed for a period of 1–2 hours, following which they were dried in the oven at 100°C for a median time of 18 hours (min 14 h–max 54 h) and cooled to room temperature for a period of 1–2 hours. After each drying period, masses of the jars were measured, and ultrapure water was added to reach 40% WHC. After 20 cycles, biochar particles were separated from the sand by wet sieving using a US mesh size no. 270 sieve that allowed the biochar particles less than 45 μm in size to pass through while retaining the sand particles.

**Chemical ageing.** For chemical ageing, we treated biochar samples with $H_2O_2$ based on the method reported by Huff and Lee [22]. We used a high concentration of $H_2O_2$ based on findings from previous studies that reported maximum changes in surface chemistry of biochar upon treatment with 30% w/w $H_2O_2$ solution [22, 40]. Briefly, 30% w/w $H_2O_2$ solution was added to 5 g of biochar at a ratio of 1 g biochar: 20 mL solution and shaken inside a chemical fume hood for 2 hours at 100 rpm. After 2 hours of shaking, we filtered the biochar samples

through sterile Whatman glass microfiber filters (Grade 934-AH Circles– 1.5 μm particle retention) and rinsed with 100 mL aliquots of ultrapure water to remove any residual $H_2O_2$.

**Biological ageing.** For biological ageing, we exposed the biochar samples to a microbial community in a nutrient solution supplemented with glucose (40 μg glucose mg$^{-1}$ biochar C), building on the method reported by Hale et al. [19]. We added glucose to stimulate microbial activity and, with it, the decomposition of biochar. We chose a microbial community expected to be enriched in microbes that could degrade biochar to further accelerate the biological ageing treatment. We derived the microbial inoculum from soil samples collected at the Blodgett Forest Research Station at University of California, Berkeley, which has been used to conduct multiple prescribed burn studies [41]. The soil samples for the inoculum were collected from 0–10 cm depth at the center of a slash pile burn after removing the ash layers. To extract the inoculum, we mixed the field-moist soil samples with Millipore water in sterile 50 mL centrifuge tubes and vortexed the tubes for 2 hours at high speed. After vortexing, the tubes were allowed to stand for 5 minutes, and the soil suspensions were filtered through sterile 2.7 μm Whatman membrane filters into sterile centrifuge tubes. For the biological ageing process, nutrient solution was prepared from autoclave-sterilized basal salt solution (500 mL L$^{-1}$ final biochar nutrient media), modified from Stevenson et al. [42], filter-sterilized vitamin B12 solution (200 μL L$^{-1}$ final biochar nutrient media), filter-sterilized vitamin mixture (200 μL L$^{-1}$ final biochar nutrient media) and a filter-sterilized trace elements solution (1 mL L$^{-1}$ final biochar nutrient media) [43]. We combined 5 g of biochar and glucose supplement (40 μg mg$^{-1}$ biochar carbon) with 250 mL ultrapure water and autoclave-sterilized the mixture. After autoclaving, the biochar mixture was transferred to a quart-sized (946.35 mL) Mason jar and combined with 250 mL of the nutrient solution. Note that the pH of the basal salt solution, which is a part of the nutrient solution was adjusted to 7 to obtain pH neutral final biochar nutrient media. The detailed composition of the nutrient solution along with the steps for preparation of the final biochar nutrient media are provided as supplementary material accompanying this work (S1 Appendix). To the resulting biochar and glucose supplemented nutrient media, we added 8 mL of the filtered inoculum and incubated the jars at 30˚C in a shaker incubator set to 100 rpm for a period of 2 weeks to allow for biological ageing to take place.

## Incubation

We performed the incubations with all the aged biochar (PHY, CHEM and BIO) as well as unaged biochar (UN) produced at both 350 and 550˚C as solid agar biochar media, inoculated with a bacterial isolate known to grow on biochar at room temperature, while tracing $CO_2$ emissions from each replicate.

The bacterial isolate we used was a *Streptomyces* that was isolated on media with eastern white pine wood biochar produced at 500˚C as the sole C source. We confirmed that the isolate was a *Streptomyces* by Sanger sequencing the full length 16S ribosomal RNA gene and BLASTing the sequences against the GenBank database. The primary motivation for selecting this specific isolate is that it was able to grow on biochar media during trial lab incubations. Further, there is evidence that indicates that bacterial genera that respond positively to biochar addition in soils include members that have the potential to break down polycyclic aromatic hydrocarbons (PAHs) [44], a constituent of biochar, particularly high-temperature ones. We recovered the isolate from glycerol stocks by streaking onto a biochar (produced from pine wood at 350˚C) nutrient media agar plate (as described in S1 Appendix) and incubating for 5 days at 37˚C. A single colony from the biochar media plate was inoculated into 30 g L$^{-1}$ tryptic soy broth (Neogen Culture Media, Lansing, MI, USA) and incubated at 30˚C in a shaking incubator until growth was visible, characterized by turbidity in the media.

We performed incubations in quarter-pint sized Mason jars (118.29 mL). The final biochar nutrient media that was used for the study was prepared by combining the nutrient solution with a biochar agar suspension. The nutrient solution was prepared as described earlier under "biological ageing" section. A suspension of biochar (1 g $L^{-1}$ final biochar nutrient media) and noble agar (30 g $L^{-1}$ final biochar nutrient media) was sterilized by autoclaving and combined with the nutrient solution to obtain a pH neutral final biochar nutrient media (S1 Appendix). For each sample, we poured 40 mL of the final biochar nutrient media into sterile Mason jars. This volume was estimated from previous incubations, where we confirmed that it was suffi-cient to ensure that the final media did not dry out during incubation. After the agar solidified, the plates were inoculated with 20 μL of the bacterial suspension in malt extract broth and plated onto the agar surface using the spread plate technique [45]. All our treatment jars received the same volume of the bacterial suspension during inoculation.

We performed the incubations in replicates of three for each treatment (five each for 350BIO and 550BIO) and included uninoculated controls for each treatment. The uninoculated controls were included to account for $CO_2$ that may accumulate due to abiotic degradation of biochar. In addition, we included two empty jars as gas flux blanks for the experiment. After plating and inoculation, the jars were capped and sealed with sterile, gas-tight lids with fittings for $CO_2$ gas measurements and attached to randomly selected positions on the distribution manifolds (multi-plexer) using polyurethane tubing [46]. We incubated the jars at room temperature and mea-sured the concentration of $CO_2$ respired in the headspace of each jar at intervals of 3–4 days using a Picarro G2131i cavity ringdown spectrometer attached to the multiplexer over a period of one month. After each measurement, we flushed the jars with a 400 ppm $CO_2$-air gas mixture to ensure aerobic conditions inside the jar. The precise concentration after flushing each jar was measured and subtracted from the next time point reading to determine the respired $CO_2$ in the jar. From previous biochar incubation trials with the isolate, we confirmed that sampling over a 3-4-day interval did not lead to oxygen depletion inside the jars.

The raw $CO_2$ readings measured using the multiplexer-Picarro system were processed in R to calculate biochar C mineralized over the period of incubation using the following packages: 'tidyverse' [47], 'zoo' [48], 'RColorBrewer' [49] and 'broom' [50]. Briefly, we calculated the cumulative biochar C mineralized for each replicate at each time point. We calculated the bio-logical cumulative biochar C mineralized values for all replicates by subtracting the corre-sponding mean C mineralized of uninoculated replicates within each treatment. The biological cumulative biochar C mineralized values were normalized by the mean biochar C content within each treatment and a time series was plotted comparing the biochar C minerali-zation trends between the aged and unaged biochar samples.

After the incubation period, we disconnected the jars and analyzed images of the agar sur-faces using the software *ImageJ* [51]. The percentage area occupied by the growth of bacterial colonies was determined for each incubation jar and used as a rough proxy to compare micro-bial growth between jars (S1 Fig).

## Chemical analyses

Total C and N were determined for aged and unaged biochar samples using a Thermo Scien-tific Flash EA 1112 Flash Combustion Analyzer (Thermo Fisher Scientific, Waltham, MA, USA) at the Department of Agronomy, UW- Madison, WI, USA. Total H was determined using a Thermo Delta V isotope ratio mass spectrometer interfaced to a Temperature Conver-sion Elemental Analyzer (Thermo Fisher Scientific, Waltham, MA, USA) at the Cornell Iso-tope Laboratory, NY, USA. Total O was calculated by subtraction as per Enders et al. [14], after determining ash content of aged and unaged biochar samples using the method

prescribed by ASTM D1762-84 Standard Test Method for Chemical Analysis of Wood Charcoal (See further details in S1 Appendix).

The pH of aged and unaged biochar samples was measured in deionized water at a 1:20 solid: solution ratio using an Inlab Micro Combination pH electrode (Mettler Toledo, Columbus, OH, USA) connected to a Thermo Scientific Orion Star A111 benchtop pH meter (Thermo Fisher Scientific, Waltham, MA, USA). Further details of this procedure can be found in the S1 Appendix.

The FT-IR measurements were performed at the U.S. Dairy Forage Research Center, Madison, WI, USA with a Shimadzu IRPrestige-21 FT-IR spectrometer (Shimadzu, Kyoto, Japan) on the ATR (Attenuated Total Reflection) absorbance mode. Briefly, 5–10 mg of the biochar sample was placed on the Zn-Se sample trough and scanned. For each sample, we obtained 256 scans per sample in the range from 4000 to 650 $cm^{-1}$ with a resolution of 1 $cm^{-1}$ (550UN, 350PHY and 550PHY) and 2 $cm^{-1}$ (350CHEM, 550CHEM, 350BIO, 550BIO and 350UN). Background corrections were performed between each sample measurement. We assigned wavenumbers for selected functional groups based on previous studies (S1 Table) and quantified the peak heights of selected functional groups after spectrum normalization using the Shimadzu IR Solution FT-IR software. Fractional signal heights for each of the FT-IR peaks were calculated by dividing the signal height of each of the peaks by the sum total of signal heights of all peaks of interest to determine the contribution of the signal generated by a particular species to the full spectra. Peaks for OH and $CO_2$ stretching at 3370 $cm^{-1}$ and 2350 $cm^{-1}$, respectively, are identified in figures but were not expected to reflect meaningful differences in PyOM chemistry, and hence were not included in our fractional signal heights calculations. Further details of this procedure can be found in the S1 Appendix.

## Statistical methods

We performed most calculations in R. Figures were made using the 'ggplot2' [52] and 'wesanderson' [53] packages. All code used for analyses and figures in this paper is available at github. com/nayelazeba/biochar-ageing.

We used the Shapiro-Wilk test to check for normality of data. Since the data did not follow a normal distribution, non-parametric tests were used to compare significant differences between cumulative biochar C mineralized. The Kruskal–Wallis one-way analysis of variance (ANOVA) and a nonparametric multiple comparison Dunn's test were used to investigate significant differences in cumulative biochar C mineralized between groups of temperature and ageing treatments. In order to determine significant correlations between cumulative biochar C mineralized and molar O/C ratios, we used the Spearman's rank correlation analysis. The above-mentioned tests were all performed using the 'stats' package in R.

To compare the full FT-IR spectra of biochar samples across temperatures and different ageing treatments, we used a multivariate dendrogram technique. We used the continuous normalized spectral data for these analyses, excluding the region from 4000 $cm^{-1}$–3100 $cm^{-1}$ wavenumber to remove signals from water sorbed to the biochar surface. We used the 'dendextend' [54] package in R to construct a dendrogram. Euclidean distances between biochar samples were calculated using the dist() function, and the hclust() function with the complete linkage method used for hierarchical clustering, where the two most similar samples are clustered together, one after another, forming an ordered hierarchical tree/ dendrogram.

## Results and discussion

### Effect of ageing on biochar C mineralization

The biochar C mineralization trends for all treatments show a similar pattern overall, with an initial period of steep increase in C mineralization, followed by the onset of a period of lower

C mineralization (about 350 hours after the start of incubation, Fig 1 and S2 Fig). The mean cumulative C mineralized in uninoculated replicates at the end of the incubation period was 0.04 mg $CO_2$-C $g^{-1}$ biochar-C for 350˚C biochar and 0.03 mg $CO_2$-C $g^{-1}$ biochar-C for 550˚C biochar. Mean cumulative biochar C mineralized at the end of the incubation was significantly higher by 39% for 350˚C biochars compared to 550˚C biochars (Kruskal-Wallis$_{ANOVA}$, $p$ value = 0.01). The difference in cumulative C mineralization between 350˚C and 550˚C biochars is further discussed in S1 Appendix.

Amongst the 350˚C chars, the mean cumulative biochar C mineralized for aged biochars was higher than that for unaged biochar through the entire incubation period (Fig 1). At the end of the incubation period, the mean cumulative biochar C mineralized was significantly

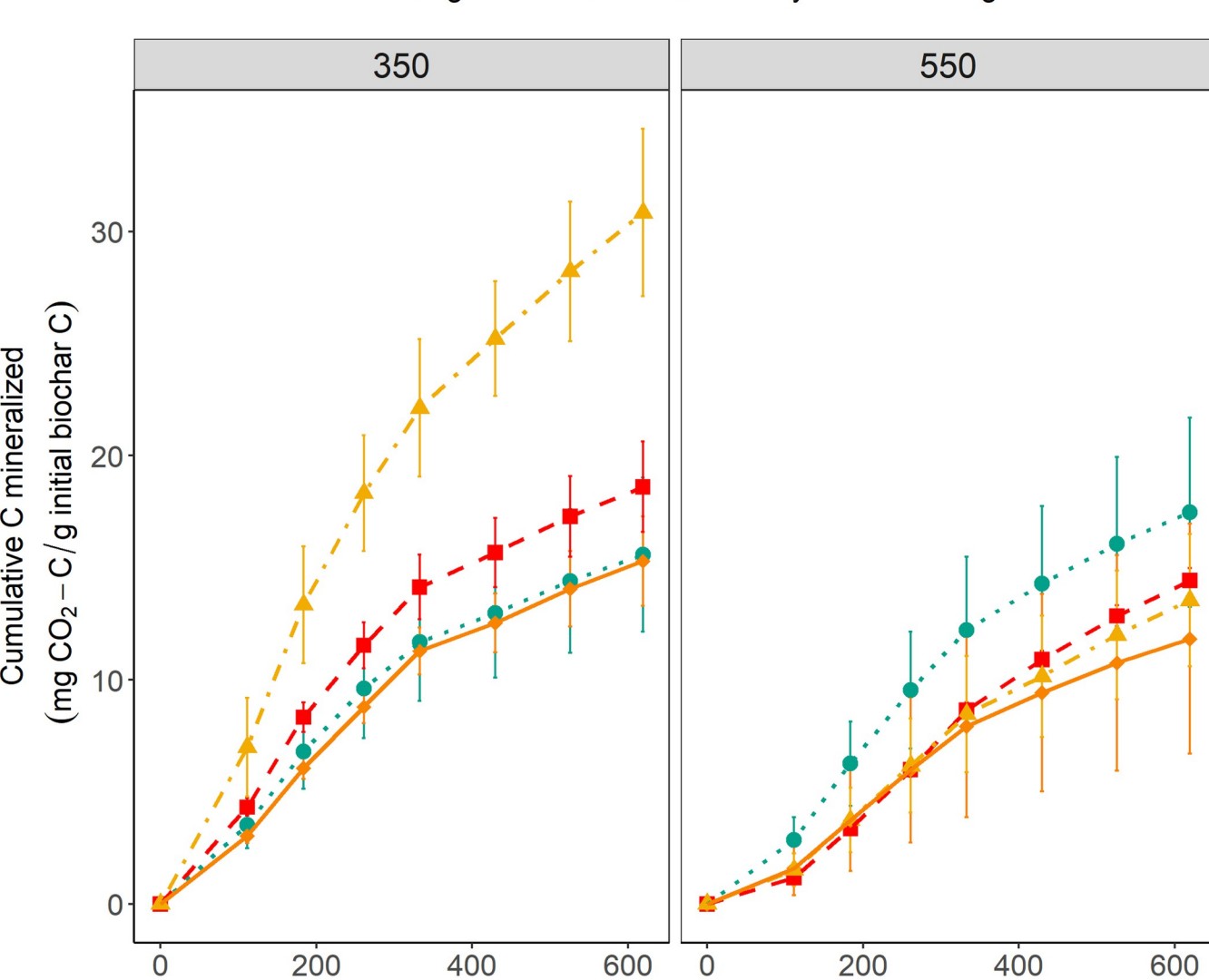

**Fig 1. Cumulative biochar C mineralization over time.** Data represent mean cumulative C mineralized from unaged and physically, chemically and biologically aged biochar samples over time, with uninoculated blanks subtracted and normalized with mean biochar-C. N = 3 for physical, chemical and unaged, N = 5 for biological. Error bars represent 95% confidence intervals. The left panel shows biochar produced at 350˚C and the right panel shows biochar produced at 550˚C.

different between the ageing treatments (Kruskal-Wallis$_{ANOVA}$, *p* value = 0.03). The greatest increase was measured for 350PHY which showed 103% higher C mineralization compared to unaged biochar, although we did not identify a statistically significant difference using the Dunn's test ($p_{adj}$ value = 0.05). The rate of C mineralization was highest for 350PHY at the onset of the incubation and remained higher than the C mineralization rate of unaged biochar throughout the incubation period (S2 Fig). 350PHY treatments also showed 87% higher surface growth of the isolate compared to unaged biochar at the end of the incubation period (S3 Fig), consistent with the cumulative biochar C mineralized data. The mean cumulative biochar C mineralization was 22% higher for 350BIO compared to the unaged treatment (Dunn's test, $p_{adj}$ value = 0.29), while the increase for 350CHEM was negligible.

Amongst the 550°C biochar, there were not large differences between mean cumulative biochar C mineralization in aged versus unaged biochar. We observed an increase in biochar C mineralized for 550CHEM compared to unaged biochar through the incubation period and a slight increase in 550PHY and 550BIO after about 400 hours after the start of incubation (Fig 1). The mean cumulative biochar C mineralized at the end of the incubation period was 47% higher for 550CHEM compared to unaged biochar and higher by 15% and 22% for 550PHY and 550BIO respectively, but the differences in means were not significant (Kruskal-Wallis$_{ANOVA}$, *p* value = 0.40). These observations were consistent with trends in growth measurements of the isolate on 550°C biochar agar surfaces, where the mean surface growth was 43.4% greater for 550CHEM compared to 550UN but no difference in growth was observed for 550BIO and 550PHY treatments (S3 Fig).

## Changes in elemental composition during ageing

Aged biochars produced at both 350°C and 550°C had lower mean total C and higher mean total O contents than unaged biochar, except in the case of 350CHEM, where we did not observe similar trends (Table 1). The molar O/C ratio increased for 350BIO (0.26) and 350PHY (0.39). The O/C ratio for the control 350UN was 0.20. In the case of 550°C chars, the O/C ratio increased for 550BIO (0.18), 550PHY (0.15) and was highest for 550CHEM (0.26) compared to 550UN (0.11). This is consistent with previous studies that have shown an increase in O/C ratio following natural as well as artificial ageing of biochar through abiotic and biotic processes [15, 20, 21, 32, 55]. The relative decrease in C with ageing is likely due in

**Table 1. Elemental composition, elemental ratio, and pH of the unaged and physically, chemically and biologically aged biochar samples produced at low temperature (350°C) and high temperature (550°C).**

| HTT (°C) | Ageing treatment | Total C | Total N | Total H | Ash | Derived total O | O/C | H/C | pH in solution |
|---|---|---|---|---|---|---|---|---|---|
| | | | | | (wt %) | | | | |
| 350 | Unaged | 75 | 0.3 | 3.9 | 0.6 | 20.4 | 0.20 | 0.62 | 6.1 |
| | Physical | 61 | 0.3 | 2.7 | 4.3 | 31.8 | 0.39 | 0.53 | 3.3[†] |
| | Chemical | 80 ± 4.6[*] | 0.3 | 2.5 | 1.6 | 15.6 | 0.15 | 0.38 | 4.3 |
| | Biological | 70 | 0.3 | 3.5 | 1.8 | 24.4 | 0.26 | 0.60 | 4.8 |
| 550 | Unaged | 85 ± 1.2[*] | 0.2 | 2.4 | 0.8 | 11.9 | 0.11 | 0.34 | 6.9 |
| | Physical | 79 | 0.4 | 2.2 | 3.1 | 15.8 | 0.15 | 0.34 | 6.5[†] |
| | Chemical | 71 | 0.3 | 3.8 | 0.7 | 24.4 | 0.26 | 0.63 | 5.0 |
| | Biological | 77 | 0.3 | 2.5 | 2.4 | 18.2 | 0.18 | 0.39 | 4.8 |

Note: Data shown represent the mean of all lab replicates unless specified otherwise.

[*]Mean ± standard deviation is shown for data where the standard deviation of lab replicates (N = 4) is > 1.

[†]No replicate measurements were included due to sample limitation.

part to leaching of C-rich dissolved organic matter [20]. Additionally, abiotic oxidation of C to carbon dioxide and utilization of C as a substrate by microbes in the case of biologically aged biochars is likely to result in relatively greater loss of C than O [10, 25, 32, 56]. The higher O content in aged biochars indicates an increase in O-containing functional groups that is likely due to both abiotic oxidation of C in the case of chemically and physically aged biochars [15, 32] as well as microbially mediated oxidation in the case of biologically aged biochar [55]. The effects of pyrolysis temperature on the elemental composition of biochar are discussed in S1 Appendix.

## Changes in surface chemistry during ageing

Amongst the ageing treatments, the surface chemistry of physically aged biochar was altered the most when compared against the control, as indicated by their spectra being most dissimilar from the unaged biochar's (Fig 2B). While chemical and biological ageing also led to changes in surface chemistry, we see these samples cluster together more by production temperature than treatment method–*i.e.*, production temperature was a more important determinant of biochar chemistry than ageing treatment. An important factor distinguishing 350˚C and 550˚C biochars is the increase in aromatic carbon content and decrease in H and O-containing functional groups on the surface of biochar with increasing pyrolysis temperatures [10, 11, 39] (See S1 Appendix for further discussion on effects of pyrolysis temperature on surface chemistry). It is interesting to note that the two PHY samples are more similar to each other than to other samples produced at the same pyrolysis temperatures. This suggests that the surface chemistry in PHY samples was more strongly affected by the ageing treatment than the production temperature. It is possible that subjecting biochar to repeated freeze-thaw-wet-dry cycles during physical ageing altered the particle structure, thereby causing an increase in the surface area over which these modifications would take place, although we did not investigate changes in surface area during ageing.

An important feature that stood out when comparing the FTIR spectra of unaged and aged biochars was the increase in O-containing carboxylic groups, measured by changes in the relative peak height of the C = O stretch at 1701 cm$^{-1}$ wavenumber (Fig 2A and S2 Table). PHY samples across pyrolysis temperatures showed the maximum values for C = O stretch, indicating that the surfaces of physically aged biochar were the most oxidized and rich in carboxylic groups. We also measured a slight increase in carboxylic groups for both 350CHEM and 550CHEM compared to unaged chars. The increase in surface oxygenation and O-containing functional groups after ageing is consistent with the findings of previous studies that investigated changes in surface chemistry using methods analogous to the physical and chemical treatments used in this study [15, 20–23]. For biological ageing, we observed a relative increase in carboxylic groups only in the case of 350BIO. This suggests that abiotic oxidation through physical and chemical ageing methods used in the study resulted in more surface oxidation and carboxylic groups compared to biotic oxidation through biological ageing. This agrees with the finding of Cheng et al. [32], where they noted that abiotic processes were more important than biotic processes for the initial surface oxidation of fresh biochar.

Amongst the surface aromatic and aliphatic groups, we consistently observed a decrease in relative peak height in the 1413 cm$^{-1}$ aliphatic C-H stretch, 810 cm$^{-1}$ aromatic C-H stretch and 1593 cm$^{-1}$ C = C aromatic stretch regions after ageing in 350˚C biochar. The maximum decrease in peak values was consistently observed for 350PHY. Additionally, in the case of 350PHY, we measured a considerable decrease in relative peak height for the aliphatic C-H stretch at 2932 cm$^{-1}$ after ageing. In the case of 550˚C char, we observed a considerable decrease in the relative peak height for the aromatic C-H stretch and a slight decrease in the

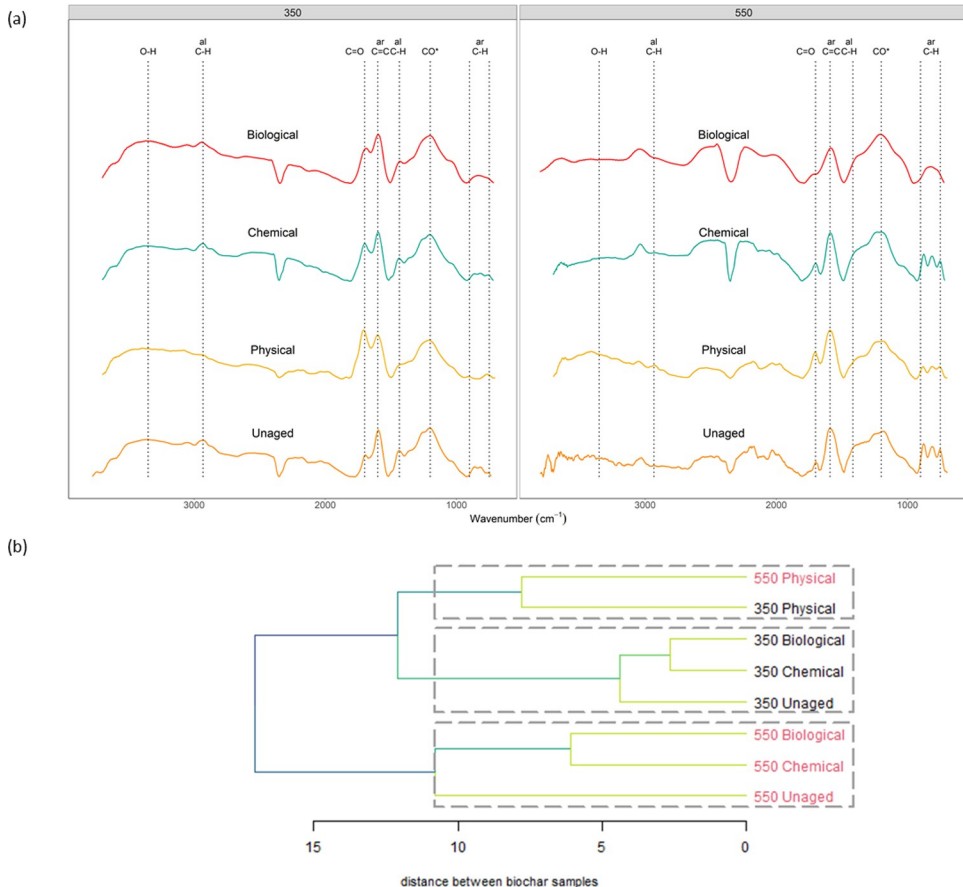

**Fig 2.** Changes in surface chemistry during ageing inferred using FTIR spectroscopy **(a)** FT-IR spectra of unaged and physically, chemically and biologically aged biochar samples produced at 350°C (left panel) and 550°C (right panel). Labels on top indicate the peak names assigned to different functional groups as described in detail in supplementary information (O-H: O–H stretching in carboxylic acids, phenols, alcohols at 3370 cm$^{-1}$; al CH: aliphatic C–H stretch in CH$_3$ and CH$_2$ at ~2932 cm$^{-1}$ and C–H bending of CH$_3$ and CH$_2$ at 1413 cm$^{-1}$; CO$_2$: CO$_2$ asymmetric stretching at 2350 cm$^{-1}$; C = O: C = O stretch in carboxylic acids and ketones at ~1701 cm$^{-1}$; ar C = C: aromatic C = C vibrations and stretching of quinones at ~1593 cm$^{-1}$; CO*: C–O stretching and O–H bending of COOH and/or C–OH stretching of polysaccharides at ~1200 cm$^{-1}$; ar C-H: aromatic C-H out of plane deformation at 810 cm$^{-1}$. **(b)** The clustering of biochar FT-IR spectra based on Ward's hierarchical clustering method represented as a dendrogram. The distance of the link between any two clusters (or samples) is a measure of the relative dissimilarity between them.

1413 cm$^{-1}$ aliphatic C-H stretch after ageing but the same was not observed in the case of the C = C aromatic stretch. These changes indicate a relative loss or transformation of both surface aliphatic and aromatic carbon groups during ageing. As discussed earlier, the loss in C could be due to leaching or abiotic oxidation of C during ageing. Further, in the case of biological ageing, the relative loss in aliphatic C group at 1413 cm$^{-1}$ and 2932 cm$^{-1}$ (for 350°C chars) could be a result of decomposition of aliphatic C by soil microbes [25, 26, 57]. While it may not be possible to conclusively determine whether oxidized functional groups were previously associated with aromatic vs. aliphatic compounds, the drop in relative heights in the aromatic regions (810 and 1593 cm$^{-1}$ wavenumbers) accompanied by a relative increase in signal for car-boxyl (1701 cm$^{-1}$) group suggests that the oxidation of aromatic C results in the development of carboxylic groups. It has been previously suggested that oxidation on the edges of the aro-matic backbone of biochar, taking place over a long period of time, could lead to the formation of negatively charged carboxyl groups [56, 58, 59]. A loss in aromatic functional groups was

documented during physical ageing of peanut straw biochar [21] and during chemical ageing of pine wood biochar [22]. More recently, Yi et al. [60] measured loss and transformation of condensed aromatic C after nine years of field ageing of high temperature bamboo and rice straw biochar. It is expected that these changes would appear more slowly in studies that rely on natural ageing compared to our study where simulated ageing was more intense than natural weathering. These previous findings support the inference that the ageing methods used in this study could have caused the disruption of aromatic carbon to form carboxylic groups.

It is important to note that FTIR spectra as produced and analyzed in this study are only semi-quantitative–*i.e.*, a doubling in peak height does not necessarily represent twice as much of the bond associated with that wavenumber. Furthermore, since replicates for ageing treatments and FTIR measurements were not included, we cannot determine whether these differences are statistically significant. However, the spectra represent an average of 265 scans on pooled and homogenized samples, and consistent responses to ageing at the two different temperatures as well as consistent temperature effects across different ageing treatments both help give us confidence in the trends observed here.

## Relationship between biochar C mineralization and chemical composition

There is a significant positive correlation between biochar C mineralization across temperature and ageing treatments and the molar O/C ratio (Spearman's correlation test, $r_s$ = 0.87, $p$ value = 0.01; Fig 3). While this indicates that the ageing treatments where we observed a relative increase in the O/C ratio tended to have increased biochar C mineralizability, pyrolysis temperature is also an important factor here: we see a stronger impact of ageing on biochar C mineralization in low temperature chars. Specifically, we saw the greatest increase in cumulative and rate of biochar C mineralization compared to unaged biochar during physical ageing (Fig 1 and S2 Fig), and 350PHY was also the treatment for which the O/C ratio is the greatest and the FTIR data shows the highest increase in carboxyl groups (Fig 2A and S2 Table). We also observed a positive correlation between biochar C mineralization and the molar H/C ratio across all temperature and ageing treatments, although, the correlation was not statistically significant (Spearman's correlation test, $r_s$ = 0.67, $p$ value = 0.08; S4 Fig). An increase in the carboxylic groups, molar O/C and molar H/C ratio of biochar during ageing could make it less stable, more hydrophilic and more likely to be mineralized by microbes [12, 61, 62]. This surface-oxidized biochar is easier to break down and could potentially facilitate the microbial metabolism of ring structures that would ordinarily be highly recalcitrant [15, 18, 56].

Additionally, in the case of 350°C biochars, we see a drop in surface aromatic groups due to ageing, with maximum drop observed in the case of 350PHY (Fig 2A and S2 Table). Oxidative transformation of aromatic C to linear alkyl-C and O-alkyl-C could decrease ring condensation and make carbon more susceptible to microbial breakdown [60]. This is observed during the incubation, after 350 hours, wherein we observed higher rate of biochar C mineralization for 350PHY than unaged biochar (S2 Fig). During this period, we would expect to see an increase in the breakdown of more complex, aromatic C by the isolate, as the easily mineralizable C is more likely to be consumed early on. Further, studies have documented breakdown and release of aromatic moieties in biochar to low molecular-weight organic acids during ageing [63, 64]. The surface oxidation of aromatic C groups has important implications for C management and cycling for both low temperature biochars and naturally produced wildfire pyrogenic organic matter (PyOM). It has been suggested that the chemical stability of PyOM produced at high temperatures during natural wildfires is more comparable to low temperature biochars produced in the lab. This is because natural PyOM was found to consist of small clusters of aromatic C units and not highly condensed polyaromatic structures [38, 65, 66].

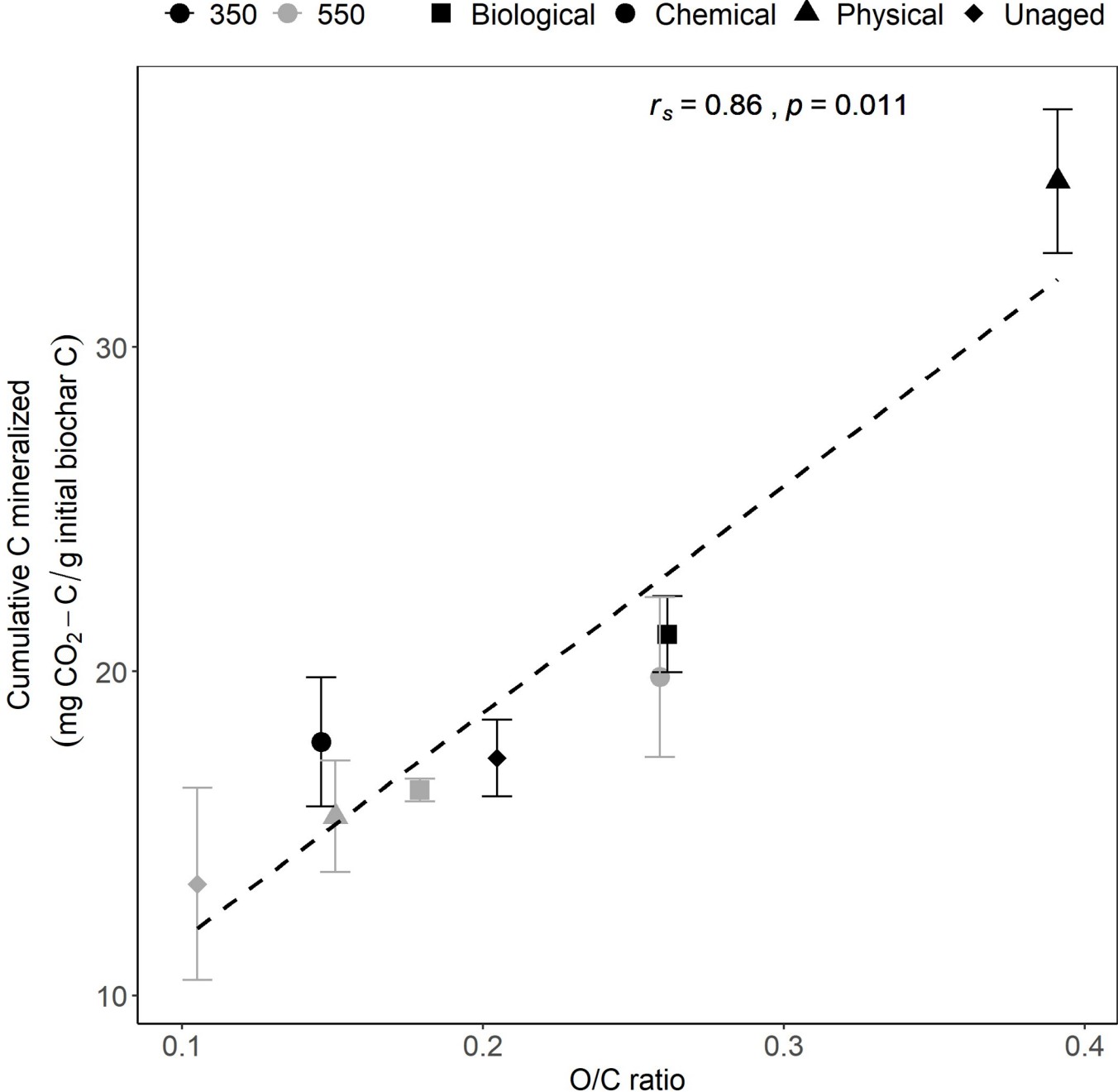

**Fig 3. Relationship between mean cumulative biochar-C mineralized and molar O/C ratio.** N = 3 for physical, chemical and unaged, N = 5 for biological treatments. Error bars represent standard error of the mean. Shapes indicate unaged, physically, chemically and biologically aged biochar samples produced at 350˚C (black) and 550˚C (gray).

Based on our findings, the carbon in these PyOM materials could be more susceptible to surface oxidation and C loss during ageing which could lead to increased mineralizability and thus decreased C storage potential.

For the 550˚C chars, we did not observe a significant increase in biochar C mineralization despite seeing an increase in the O/C and H/C ratio during ageing. This is perhaps because even though increases in surface oxidation and loss of some surface aromatic and aliphatic C

groups were observed during ageing, 550°C aged chars still retained more aromatic C and were less oxidized than 350°C aged chars, as observed in FTIR spectra (Fig 2A and S2 Table). As a result, there is likely to be less easily mineralizable C present in 550°C biochars even after ageing. This is in line with other studies that have documented an inverse relationship between mineralization and aromatic fraction in biochars [56, 67]. This effect is potentially the primary reason we did not observe a significant increase in biochar C mineralized for 550°C chars during ageing despite observing changes in the surface oxidation (Fig 1 & Fig 2A).

### Ageing considerations and future directions

The primary goal of this study is to examine the effect that long-term ageing-related changes in surface chemistry and elemental composition have on mineralizability of biochar C. To this effect, all three ageing treatments used in this study were designed to simulate extreme degrees of real-world processes that occur naturally to biochar in soil. However, it is not feasible to develop a scale to quantify the relative severity of the treatments compared to their expected severity in nature, or even measure the relative severity of the treatments. It is possible that further increasing the severity of the biological and chemical ageing processes could lead to increased C mineralization for those treatments as well. For instance, increasing the duration of biological ageing may result in a continued increase in biochar surface oxidation and O/C ratio. Despite the lack of quantitative comparison, this study highlights the increased susceptibility of biochar to microbial degradation with increase in surface oxidation. In particular, it demonstrates the ability of abiotic processes such as freeze-thaw-wet-dry cycles in accelerating surface oxidation. The precise mechanism by which oxidation of biochar takes place during physical ageing remains unknown and warrants further research [30].

It is important to note that ageing and incubation of biochar was performed in the absence of soil. However, we note that in soil systems, biochar-clay interactions, biochar-soil organic matter interactions, as well as physical protection of biochar through aggregate formation are likely to affect both ageing of biochar and its interactions with microbes [68–70]. Further investigation into changes in bulk and surface properties associated with long term ageing of biochar in biochar amended soils could help in verification of laboratory biochar ageing and incubation studies as well as broaden our understanding of the potential of biochar as a C sink [30]. While this study focusses on changes in surface chemistry and elemental composition, we note that other effects of ageing such as changes in specific surface area, cation exchange capacity and solubility will affect how microbes interact with biochar in soil, in terms of accessibility to the biochar as well as its mineralizability.

We note that despite this *Streptomyces* isolate being a known degrader of unaged PyOM, we still observed a trend of increase in biochar C mineralization after ageing, particularly in the case of physical ageing. For other soil microbes that do not share this capability of decomposing aromatic C in PyOM, we would predict ageing could potentially have a larger impact on their ability to break down PyOM. This would be interesting to test directly, as it could have implications in how post fire C is cycled by microbes based on their affinity for metabolizing PyOM. It would also be interesting to perform future experiments on PyOM degradation using consortia of microbes, in addition to single isolates.

In conclusion, this study provides evidence that higher O/C ratio and surface oxidation during ageing is likely to accelerate biochar C mineralization by microbes. The greatest relative increase in surface carboxylic groups and O/C ratio was observed during physical ageing in low temperature char. As such, our lab experiments demonstrate that the C in low temperature biochar is more susceptible to oxidation and microbial mineralization. Further research is

required to better understand how ageing due to both abiotic and microbial processes in soil will affect the C sequestration potential of biochar.

## Supporting information

**S1 Table. FT-IR functional group peak assignments for biochar.**
(DOCX)

**S2 Table. FTIR spectra relative peak heights of the unaged and physically, chemically and biologically aged biochar samples produced at low temperature (350˚C) and high temperature (550˚C).**
(DOCX)

**S1 Fig. Images of *Streptomyces* isolate growth on biochar- raw and processed using *ImageJ*.**
(Left) Images of Streptomyces isolate growth on the surface of biochar nutrient agar media at the end of the incubation period for a replicate of (a) 350˚C unaged biochar (b) 550˚C unaged biochar (c) 350˚C physically aged biochar (d) 550˚C chemically aged biochar. (Right) Processed images of Streptomyces isolate growth on the surface of the biochar sample shown on the left using the *ImageJ* software.
(DOCX)

**S2 Fig. Rate of biochar C mineralization over time.** Data show mean C mineralization rate of unaged and physically, chemically and biologically aged biochar samples over time, with uninoculated blanks subtracted and normalized with mean biochar-C. N = 3 for physical, chemical and unaged, N = 5 for biological. Error bars represent 95% confidence intervals. The left panel shows biochar produced at 350˚C and the right panel shows biochar produced at 550˚C.
(TIF)

**S3 Fig. Comparison of *Streptomyces* isolate growth on aged and unaged biochar nutrient agar media.** Growth of Streptomyces isolate on biologically, chemically, and physically, aged biochar and unaged biochar agar media over the incubation period. N = 3 for physical, chemical and unaged, N = 5 for biological. The left panel shows biochars produced at 350˚C and the right panel shows biochars produced at 550˚C.
(TIF)

**S4 Fig. Relationship between mean cumulative biochar-C mineralized and molar H/C ratio.** N = 3 for physical, chemical and unaged, N = 5 for biological treatments. Error bars represent standard error of the mean. Shapes indicate unaged, physically, chemically and biologically aged biochar samples produced at 350˚C (black) and 550˚C (gray).
(TIF)

**S1 Appendix. Biochar production and chemical analyses, biochar nutrient media preparation, effect of pyrolysis temperature on biochar C mineralization of unaged biochar, effect of pyrolysis temperature on the elemental composition and surface chemistry of unaged biochar.**
(DOCX)

## Acknowledgments

We thank Akio Enders for supplying the biochar that we used to isolate *Streptomyces* and for assistance with biochar production. We are thankful to Maggie Phillips at the Jackson Lab and Kim Sparks at the Cornell Stable Isotope Laboratory for assistance with chemical analyses.

Thanks to Jamie Woolet for isolating *Streptomyces* on biochar and for all the technical assistance with the incubations. We also thank Monika Fischer and Neem Patel for supplying the soil samples that we used during biological ageing.

## Author Contributions

**Conceptualization:** Nayela Zeba, Thea Whitman.

**Formal analysis:** Nayela Zeba, Timothy D. Berry, Kevin Panke-Buisse, Thea Whitman.

**Funding acquisition:** Thea Whitman.

**Investigation:** Nayela Zeba, Timothy D. Berry, Kevin Panke-Buisse.

**Supervision:** Thea Whitman.

**Writing – original draft:** Nayela Zeba.

**Writing – review & editing:** Nayela Zeba, Timothy D. Berry, Kevin Panke-Buisse, Thea Whitman.

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
