## [Decision Letter · Decision Letter 0]

6 Jan 2022

PONE-D-21-38050Effects of physical, chemical, and biological ageing on the mineralization of pine wood biochar by a Streptomyces isolatePLOS ONE

Dear Dr. Whitman,

Thank you for submitting your manuscript to PLOS ONE. After careful consideration, we feel that it has merit but does not fully meet PLOS ONE’s publication criteria as it currently stands. Therefore, we invite you to submit a revised version of the manuscript that addresses the points raised during the review process.

ACADEMIC EDITOR:

We look forward to receiving your revised manuscript.

Kind regards,

Omeid Rahmani

Academic Editor

PLOS ONE

Journal Requirements:

[This research was funded by the U.S. Department of Energy (DE-SC0020351; DE-SC0016365). We thank Akio Enders for supplying the biochar that we used to isolate Streptomyces and for assistance with biochar production. We are thankful to Maggie Phillips at the Jackson Lab and Kim Sparks at the Cornell Stable Isotope Laboratory for assistance with chemical analyses. Thanks to Jamie Woolet for isolating Streptomyces on biochar and for all the technical assistance with the incubations. We also thank Monika Fischer and Neem Patel for supplying the soil samples that we used during biological ageing.]

 [TW: DE-SC0020351 and DE-SC0016365, DOE Office of Science, Office of Biological and Environmental Research (BER), https://science.osti.gov/ber/. The funders had no role in study design, data collection and analysis, decision to publish, or preparation of the manuscript.]

Reviewers' comments:

Reviewer's Responses to Questions

**Comments to the Author**

1. Is the manuscript technically sound, and do the data support the conclusions?

Reviewer #1: Yes

2. Has the statistical analysis been performed appropriately and rigorously? 

Reviewer #1: Yes

3. Have the authors made all data underlying the findings in their manuscript fully available?

Reviewer #1: Yes

4. Is the manuscript presented in an intelligible fashion and written in standard English?

Reviewer #1: Yes

5. Review Comments to the Author

Reviewer #1: Overall

Nice study. Appropriate methods and controls used. Written up clearly. Almost publishable as is but could use a more developed discussion of hypothetical mechanisms at play and real world implications.

Abstract

-Abstract could use clarification as to the differences between phys, chem, and biol aging on C mineralizartion

-Ln23: no need for “suggesting increased mineralizability of aged biochar C.”

-Ln27: What ‘implications’?

-comment on whether the presence of the microbe had an effect

Intro.

-Ln36: Confusing. “particularly the baseline scenario for the fate of the parent biomass” Be more specific

-Ln71: tranches is an odd word to use here

-Ln 76: Explain why ‘increase in O content (as indicated by O/C ratios), would lead to higher mineralization

-Is there preliminary research suggesting that chem vs phys vs bio aging will have different effects?

-Ln70-77: would be more clear if you were perhaps hypothesizing difference due to skeletal vs functional group chemistry

Methods

-Ln84: You mean peak T residence time?

-Ln97-102: All this belongs in Intro section or is redundant with Introduction

-Is there a reference that can be added (or SI) to indicate that it Streptomyces was able to grow on biochar media during trial lab incubations? How did you know you were using a Streptomyces species?

-Ln183: Make more clear : “final biochar nutrient media” and “Noble agar suspension (30 g L-1 final biochar nutrient media)” , not just in SI

Results/Discussion

-Ln288-289 (and throughout discussion): The question is not whether mineralized C was higher throughout the incubation period, but was incubation RATE always higher – rates were clearly higher for aged biochar n the early period, but what about later (i.e. after about 400 h)?

-Ln311: ‘growth refers to what?

-First 2 sentences of section 3.3 say the same thing

-I think the authors should look more into the reasons why phys aging produced the largest chemical and lability effects. Did other studies find similar results?

-Ln430: It could also be that oxidation makes biochar more soluble and this is what allows mineralization. See solubility and dissolved biochar studies. Solubility limitation could also explain lack of 550 C increase in min with aging.

-Would be interesting to compare rates of biochar C mineralization in this study (using Streptomyces) with other previous studies that used microbial consortia

-Ln474: I don’t understand ‘not compare between the various treatments.’

-section ‘3.5. Ageing considerations and future directions’ is rather mundane. Instead, discuss implication in nature and global C cycle. Also, Further research directions should be more specific

Figs/Tabs

- I hate having tables and figure captions placed inside of the text rather than all together at the end

-Please include abiotic control treatments in fig 1 (or at least discuss).

-either a Fig or a Table of FTIR data should be in the main manuscript, not both

-What was clustered in Fig. 2b?

6. PLOS authors have the option to publish the peer review history of their article (what does this mean?). If published, this will include your full peer review and any attached files.

Reviewer #1: No

---

## [Author Response · Author response to Decision Letter 0]

18 Feb 2022

Please see attached response to reviewers document. We have also copied the text of this document here:

Reviewer #1: Overall

Nice study. Appropriate methods and controls used. Written up clearly. Almost publishable as is but could use a more developed discussion of hypothetical mechanisms at play and real world implications.

We thank the reviewer for their thoughtful comments and constructive feedback. We have addressed the reviewer’s comments below and made the required changes to the manuscript.

Abstract

-Abstract could use clarification as to the differences between phys, chem, and biol aging on C mineralizartion

We have revised lines 15-18 to emphasise the differences between physical, chemical and biological ageing.

-Ln23: no need for “suggesting increased mineralizability of aged biochar C.”

We agree- this part has been deleted.

-Ln27: What ‘implications’?

We have included a sentence in the abstract in line 28 that spells out how our finding that 350 °C biochar is more susceptible to mineralization by soil microbes has implications for long term stability of low temperature biochar in soil. The implications have been discussed in the Results and discussion section “Relationship between biochar C mineralization and chemical composition” in lines 424-432. 

-comment on whether the presence of the microbe had an effect

We assume that the reviewer is referring to the biological ageing treatment wherein we aged biochar by incubating it with a microbial community extracted from soil. Biological ageing did result in an increase in O/C ratio for 350BIO and 550BIO (discussed in lines 306-309) and an increase in surface carboxylic groups for 350BIO (discussed in lines 358-363). However, these changes did not result in a significant increase in mineralization for 350 °C and 550 °C biologically aged biochar. If the reviewer is thinking of whether the isolate had an effect on the biochar, we did not characterize the char for a third time after the C mineralization experiment.

Intro.

-Ln36: Confusing. “particularly the baseline scenario for the fate of the parent biomass” Be more specific

We have specified in more detail in lines 38-41 what we mean by the baseline scenario (I.e., what would have happened had it not been used to produce biochar). 

-Ln71: tranches is an odd word to use here

It is perhaps not a common word, but it is used appropriately here, in the sense of “a portion of something”.

-Ln 76: Explain why ‘increase in O content (as indicated by O/C ratios), would lead to higher mineralization

As mentioned in line 44, previous studies have established an inverse correlation between the O/C ratio and biochar persistence. This suggests that the O/C ratio is a good indicator of biochar stability. 

-Is there preliminary research suggesting that chem vs phys vs bio aging will have different effects?

The effects of chem, physical and biological ageing on the properties of biochar have been well documented in previous studies including those by Hale et al. and Mia et al. (referenced in the manuscript). The study by Hale et al. directly compares the effects of physical, chemical and biological ageing on biochar properties, specifically sorption of pyrene. However, to the best of our knowledge, the impact of ageing on biochar C mineralization has not directly been tested or compared between treatments. We highlight this in lines 58-64.

-Ln70-77: would be more clear if you were perhaps hypothesizing difference due to skeletal vs functional group chemistry

We would like to clarify that our hypothesis is not based on changes observed specifically in skeletal or functional group chemistry. With the current set of analyses, we don’t think we have a way to differentiate between the two. For instance, the increase in the elemental O content is due to the increase in O-containing functional groups on the surface. This will be reflected through FTIR measurements as well as increase in the bulk O content. Rather, our hypothesis is based on two processes that have been well documented in literature -

(i) a loss of aliphatic or easily mineralizable C during ageing, which could suggest that the aged biochar has a higher proportion of the aromatic or condensed carbon that would be difficult to decompose.

(ii) an increase in bulk O content and the O/C ratio, which would suggest that the aged biochar is easier to mineralize based on the inverse relationship between the O/C ratio and biochar stability.

Methods

-Ln84: You mean peak T residence time?

Yes, we have clarified this in line 85 and thank the reviewer for bringing this to our attention.

-Ln97-102: All this belongs in Intro section or is redundant with Introduction

We agree with the reviewer. The redundant section has been deleted.

-Is there a reference that can be added (or SI) to indicate that it Streptomyces was able to grow on biochar media during trial lab incubations? How did you know you were using a Streptomyces species?

We have provided images of growth of the Streptomyces isolate on the biochar media in the S.I. Fig. S1. These images were taken at the end of our incubation for the study presented here and provide clear evidence of the growth of the isolate. We observed the same during our trial incubations that led us to proceed with the experiment. We confirmed that the isolate was a Streptomyces species by sequencing the full length 16S rRNA gene using Sanger sequencing. This information has been added in lines 161-163 and we thank the reviewer for bringing this to our attention. 

-Ln183: Make more clear : “final biochar nutrient media” and “Noble agar suspension (30 g L-1 final biochar nutrient media)” , not just in SI

We have added an additional sentence in line 173 mentioning the composition of the final biochar nutrient media. The composition and details of the media have been described earlier in lines 141-145 where the media is first mentioned. We have also added more information to Appendix S1 in the S.I. We hope these changes bring more clarity to the composition of our media.

Results/Discussion

-Ln288-289 (and throughout discussion): The question is not whether mineralized C was higher throughout the incubation period, but was incubation RATE always higher – rates were clearly higher for aged biochar n the early period, but what about later (i.e. after about 400 h)?

This is an interesting point, and we thank the reviewer for bringing this to our attention. For the 350 °C char, we did observe higher C mineralization rates for aged biochar compared to unaged biochar within 180 hours after the start of the incubation. After 180 hours, the C mineralization decreased for both aged and unaged 350 biochar and reached a steady state after approx. 350 hours. After 350 hours, the C mineralization rate for CHEM and BIO aged biochar were similar to that of unaged 350 °C biochar, but for PHY aged biochar we continued to measure significantly higher C mineralization rates. During this period, we might expect the breakdown of more complex, aromatic C by the isolate, as the easily mineralizable C is more likely to be consumed early on. This could support the idea that physical ageing of 350 °C char likely increased the breakdown of the more aliphatic easily mineralizable carbon as well as the breakdown of the more aromatic condensed C. This is consistent with our discussion about changes to aromatic C during physical ageing and its implications in post fire soils in lines 424- 432). For the 550 °C biochar, the rate of C mineralization for all unaged biochar was higher than that of aged biochar after 350 hours of incubation but the differences were not significant. The figure showing the rate of biochar C mineralization (Fig. S2) has been added to the S.I. and a discussion about the rate of C mineralization has been added in line 278 and lines 419-423. 

-Ln311: ‘growth refers to what?

Growth of the isolate on the biochar media at the end of the incubation study as inferred from the image processing software ImageJ, described in “Materials and methods” lines 208-211. Lines 280 and 299 have been revised to include “growth of the isolate” for clarity.

-First 2 sentences of section 3.3 say the same thing

We have rephrased the opening of the section “Ageing of biochar”. It now reads, “Amongst the ageing treatments, the surface chemistry of physically aged biochar was altered the most when compared against the control, as indicated by their spectra being most dissimilar from the unaged biochar’s (Fig. 2b)."

-I think the authors should look more into the reasons why phys aging produced the largest chemical and lability effects. Did other studies find similar results?

The mechanisms by which physical ageing causes an increase in O/C ratio are certainly understudied. While a detailed investigation of these mechanisms is beyond the scope of this study, it is a topic we believe warrants further research. 

Data on chemical changes, specifically changes in elemental composition and surface chemistry resulting from physical ageing of biochar, is also inconsistent in the literature. Some studies have reported an increase in surface O content while others have reported very little or no increase in the O content following physical ageing. A recent review article by Wang et al. discusses this in more detail. To the best of our knowledge, the effects of physical ageing of biochar on mineralization have not been reported before. 

-Ln430: It could also be that oxidation makes biochar more soluble and this is what allows mineralization. See solubility and dissolved biochar studies. Solubility limitation could also explain the lack of 550 C increase in min with aging.

While it is true that oxidation during ageing increases the solubility of biochar, all biochar used in our study for the incubation was mixed with agar and ultrapure water in a suspension and autoclave sterilised before inoculation. Therefore, we do not think that solubility changes due to ageing will have a large impact on our findings.

-Would be interesting to compare rates of biochar C mineralization in this study (using Streptomyces) with other previous studies that used microbial consortia.

Since our study is focused on understanding how soil microbes might mineralize aged biochar, it would be indeed interesting to compare our findings with C mineralization data with a microbial consortium. However, there are several challenges in doing so. To the best of our knowledge, previous studies that used microbial consortia do not make an attempt to differentiate between SOC mineralization and biochar C mineralization, as they were performed directly in soil, whereas ours focuses primarily on biochar C mineralization. Further, owing to the fact that they were field studies, several confounding environmental factors make it impossible to reasonably compare absolute mineralization rates against our controlled setup. 

The best way to compare biochar C mineralization between the Streptomyces isolate and a mixed community would be to perform the same controlled experiment with a consortium extracted from soil. This is a great idea for the next step that we would like to include in our future experiments. We have added this idea to our “Ageing considerations and Future Directions” section, lines 474-481.

-Ln474: I don’t understand ‘not compare between the various treatments.’

The phrase “not compare between the various treatments” has been removed. The intention here was to clarify that it is not a goal of this study to directly compare between treatments, e.g., comparing the relative effects of say, physical vs chemical ageing. 

-section ‘3.5. Ageing considerations and future directions’ is rather mundane. Instead, discuss implication in nature and global C cycle. Also, Further research directions should be more specific

We have revised lines 463-473 to highlight the implications of ageing related changes in biochar properties in soil systems. The implications of our findings as it pertains to C stability of PyOM in post fire soil environments is discussed in lines 424-432 in the previous section “Relationship between biochar C mineralization and chemical composition”. We have also added other further research directions, particularly the idea to focus on consortia, in addition to other isolates (lines 474-481). 

Figs/Tabs

- I hate having tables and figure captions placed inside of the text rather than all together at the end

Our apologies. However, these are the requirements of PLOS One Author Guidelines: 

• Figure captions are inserted immediately after the first paragraph in which the figure is cited. Figure files are uploaded separately.

• Tables are inserted immediately after the first paragraph in which they are cited.

-Please include abiotic control treatments in fig 1 (or at least discuss).

We would like to clarify that the data points for each treatment shown in Fig. 1 have been corrected for abiotic C using their respective uninoculated treatment controls as mentioned in “Materials and methods” lines 201-207. We have now also added the cumulative mean C mineralization values for 350 and 550 °C biochar abiotic controls in “Results and discussion” (lines 265-267).

-either a Fig or a Table of FTIR data should be in the main manuscript, not both

Table 2 has been moved to the S.I. (S2 Table).

-What was clustered in Fig. 2b?

For Fig. 2b, we normalised the raw FTIR absorption spectra obtained from our samples (excluding the region from 4000 cm-1- 3100 cm-1 wavenumber) and subjected it to Ward’s hierarchical clustering analysis.

---

## [Decision Letter · Decision Letter 1]

7 Mar 2022

Effects of physical, chemical, and biological ageing on the mineralization of pine wood biochar by a Streptomyces isolate

PONE-D-21-38050R1

Dear Dr. Whitman,

We’re pleased to inform you that your manuscript has been judged scientifically suitable for publication and will be formally accepted for publication once it meets all outstanding technical requirements.

Kind regards,

Omeid Rahmani

Academic Editor

PLOS ONE

Additional Editor Comments (optional):

Reviewers' comments:

Reviewer's Responses to Questions

**Comments to the Author**

1. If the authors have adequately addressed your comments raised in a previous round of review and you feel that this manuscript is now acceptable for publication, you may indicate that here to bypass the “Comments to the Author” section, enter your conflict of interest statement in the “Confidential to Editor” section, and submit your "Accept" recommendation.

Reviewer #1: All comments have been addressed

2. Is the manuscript technically sound, and do the data support the conclusions?

Reviewer #1: Yes

3. Has the statistical analysis been performed appropriately and rigorously? 

Reviewer #1: Yes

4. Have the authors made all data underlying the findings in their manuscript fully available?

Reviewer #1: Yes

5. Is the manuscript presented in an intelligible fashion and written in standard English?

Reviewer #1: Yes

6. Review Comments to the Author

Reviewer #1: Thanks for addressing all the comments - Given that this is a re-review of a paper that required only minor revision, I dont know why this box requires a minimum character count of 100, but now I have met it

7. PLOS authors have the option to publish the peer review history of their article (what does this mean?). If published, this will include your full peer review and any attached files.

Reviewer #1: **Yes: **Andrew R Zimmerman

---

## [Editor Report · Acceptance letter]

30 Mar 2022

PONE-D-21-38050R1 

­­­Effects of physical, chemical, and biological ageing on the mineralization of pine wood biochar by a *Streptomyces* isolate

Dear Dr. Whitman:

I'm pleased to inform you that your manuscript has been deemed suitable for publication in PLOS ONE. Congratulations! Your manuscript is now with our production department. 

Kind regards, 

on behalf of

Dr. Omeid Rahmani 

Academic Editor

PLOS ONE